# Effectiveness of the Pfizer-BioNTech (BNT162b2) vaccine against the omicron variant of SARS-CoV-2 among adults aged 50 and above: A case-control study in Lebanon, June 2022

Maryo Baakliny[1,2], Nada Ghosn[2], Nadine Saleh[3,4], Patrick Maison[1]*

1 Université Paris-Est Créteil Val de Marne, Créteil, France, 2 Epidemiological Surveillance Unit, Ministry of Public Health, Beirut, Lebanon, 3 INSPECT-LB (National Institute of Public Health, Clinical Epidemiology, and Toxicology-Lebanon), Beirut, Lebanon, 4 Department of Epidemiology and Biostatistics, Faculty of Public Health, Lebanese University, Fanar, Lebanon

* patrick.maison@ansm.sante.fr

## Abstract

The emergence of the novel severe acute respiratory syndrome coronavirus 2 (SARS-CoV-2) and the subsequent coronavirus disease 2019 (COVID-19) pandemic with the continuous evolution of the virus into variants such as Alpha, Beta, Delta, Omicron and others necessitates an on-going surveillance and evaluation of vaccine effectiveness (VE).This study focuses on assessing the real-world performance of the Pfizer-BioNTech (BNT162b2) vaccine against the Omicron variant. This study employs a test-negative, case-control design, a methodology commonly utilized for estimating the VE in influenza studies. It evaluates SARS-CoV-2 test results in individuals aged 50 and older who had influenza-like illness (ILI) or COVID-like illness (CLI) symptoms and presented to the sentinel sites, classifying positive cases as "cases" and negative as "controls." Data collection was done through a standardized questionnaire administered by the Epidemiological Surveillance Unit (ESU) COVID-19 team via phone interviews. Logistic regression analysis was performed to measure the association, taking into account all confounding factors. The results indicate a 14% VE against contracting the disease among fully vaccinated individuals. Factors such as age and underlying conditions significantly influenced VE. The findings of this study highlight the effectiveness of the Pfizer-BioNTech (BNT162b2) vaccine in reducing the odds of symptomatic COVID-19, particularly by decreasing the prevalence of key symptoms such as fever, cough, myalgia, and loss of taste or smell. Breakthrough infections still occurred, indicating that while the vaccine reduces symptom severity, it may not fully prevent infection This VE is lower compared to previous variants, indicating potential challenges in vaccine efficacy. The study underscores the need for an on-going monitoring and potential booster doses to enhance protection, especially against evolving variants like Omicron.

**Data availability statement:** All data files are available from the figshare database: 10.6084/ m9.figshare.27619779

**Funding:** The author(s) received no specific funding for this work.

**Competing interests:** NO authors have competing interests in all the study.

## Introduction

In December 2019, a novel severe acute respiratory syndrome coronavirus 2 (SARS-CoV-2), which causes coronavirus disease 2019 (COVID-19), emerged. On 11 March 2020, the World Health Organization (WHO) declared COVID-19 as a pandemic [1]. SARS-CoV-2 has altered continuously into several variants; these variants include Alpha, Beta, Delta and Omicron [2].

In addition to surveillance and quarantine measures, vaccination plays a key role in the containment of viruses and pandemics. The WHO has advised that the countries with existing hospital-based sentinel influenza surveillance systems should adapt them to also monitor severe SARS-CoV-2 cases and collect data to measure COVID-19 Vaccine Effectiveness (VE) [3]. This recommendation has been implemented successfully in various countries, including the United Kingdom (UK), Egypt, India, South Africa, Lebanon and many more, demonstrating its global relevance and applicability [4].

The COVID-19 pandemic began in Wuhan, China, with an initial report of twenty-seven cases of pneumonia [5]. As of 13 February 2022, globally, there were 703 million cases infected and 6.9 million deaths [6]. Simultaneously, Lebanon reported 1.2 million cases and 10992 deaths [7]. The omicron wave started in late November 2021 in South Africa [8]. Globally, from 22 November 2021 till 23 July 2022, there were 307.76 million confirmed cases and 1.2 million deaths. While in Lebanon, 490.21 confirmed cases were registered and 1,871 deaths [7] for the same period of time.

Vaccine Effectiveness (VE) studies assess how well the vaccination against SARS-CoV-2 impacts the real-time population [9]. Throughout the world as of 17 February 2022, 10508 million vaccine doses had been administered [10]. By the same time period, a total of 5.779 million vaccine doses had been administered in Lebanon [7]. The Pfizer-BioNTech (BNT162b2) vaccine was the most administered with 4.253 million doses (73.58%) followed by AstraZeneca 0.717 million doses (12.41%); added together they make 85.99% of the total brands administered.

Evaluating the real-world performance of COVID-19 vaccines is crucial, especially given the evolving nature of the pandemic and the emergence of new variants. This study aims to assess the effectiveness of the Pfizer-BioNTech (BNT162b2) vaccine against the Omicron variant of SARS-CoV-2, focusing on individuals aged 50 and above who present influenza-like illness at sentinel sites as well as potential modifying factors. This research is significant not only for its immediate implications but also for contributing to the global understanding of vaccine effectiveness in the face of emerging viral threats. Moreover, as we move forward, the findings and methodologies from this study will serve as a valuable blueprint for researchers, policymakers, and healthcare professionals grappling with future pandemics or newly emerging viral strains. The knowledge gained here will enable more rapid and precise responses, ultimately safeguarding public health on a global scale.

## Methodology

### Study design

A test-negative case-control design was employed to calculate the odds ratios for testing positive for SARS-CoV-2 among persons who meet the standard influenza-like illness (ILI) or COVID-like illness (CLI) case definition with vaccinated (cases) versus unvaccinated (controls) individuals, all of whom were tested using polymerase chain reaction (PCR). This study design has been used in the past 10 years for estimating annual influenza VE.

### Study population

This study was conducted by recruiting cases and controls from the sentinel surveillance system data base. In the past, after the 2009 influenza A (H1N1) pandemic, many countries

implemented influenza sentinel surveillance for severe acute respiratory infections (SARI). It started in Lebanon at the end of 2014 with a main objective of monitoring influenza virus circulation by time and place [11]. The study population for this VE study was consisted of individuals aged 50 and above fitting the ILI/CLI case definition because the Pfizer-BioNTech (BNT162b2) vaccine was primarily administered to healthcare workers and individuals in this age group. 15 sites were included: (one emergency room, 3 Unrwa clinics and 11 primary health centers). The minimum number of cases and controls to detect a specified VE is 346 if the estimated vaccination coverage in the population under evaluation is around 50%, with 1–1 controls per case, with a precision of ± 10% and a type 1 error rate of 0.05 as per the interim guidance shared by WHO for VE studies [3] (S1 Fig).

### Case–controls definition

As per the WHO definition an acute respiratory infection is a case with a measured fever of ≥ 38 °C and cough; with onset within the last 10 days. Few adjustments were made to include CLI cases having anosmia or any three of the following symptoms: fever, cough, dyspnoea, myalgia, fatigue, coryza, diarrhea, nausea, anorexia, vomiting, headache, sore throat, altered mental status. Those who tested positive were considered as cases and those who tested negative as controls.

Another classification was used in the output analysis as per the Centers for Disease Control and Prevention (CDC) criteria to distinguish between mild disease, who is a symptomatic case meeting the case definition for COVID-19 without evidence of viral pneumonia or hypoxia, and moderate disease, who has the clinical signs of pneumonia (fever, with cough or dyspnoea, fast breathing).

### Definition of vaccination status

An individual will be considered as fully vaccinated if s/he had received two doses of Pfizer-BioNTech (BNT162b2) at least 14 days before the onset of symptoms. A patient will be considered as not fully vaccinated if s/he has not received COVID-19 vaccine, has received only one dose, or has been vaccinated after the onset of symptoms. The sources of information were only the documented vaccination status sent by the patients to the Ministry of Public Health (MoPH) Whatsapp.

### Data collection

The collected information was through a standardised questionnaire (S1 Table). Data collection was conducted from February 2022 till the end of April 2022. All data were entered directly on the district health information system 2 (DHIS 2), and interviews were conducted via phone by public health officers and nurses from the ESU COVID-19 team.

### Data analysis

The outcome of interest for the primary analysis was the detection of SARS-CoV-2 in patients eligible for vaccination and presented to ILI/CLI sites with symptoms fitting the case definition. Data analysis was done using R version 4.0.2 and R Studio. Descriptive statistics were calculated to summarize the demographic and clinical characteristics of the participants, frequencies and numbers were reported for categorical variables, confidence intervals (CIs) were calculated using 95% confidence levels and comparisons of percentages between groups were made using Pearson's Chi-square tests for categorical variables to assess the significance of observed differences. The measure of association used was the odds ratio (OR), which was estimated using logistic regression. This method considered all potential confounding factors

identified in the literature, along with factors that had a p-value of less than 0.2 in the bivariate analysis. Vaccine effectiveness (VE) was calculated using the formula $(1 − OR) \times 100\%$ for the ORs related to 2 doses compared to the unvaccinated group.

### Ethical considerations

No IRB approval was required as this study was part of a national surveillance system. Approval was waived under memo number 712/esu by the department committee, given that the data was collected as part of public health surveillance efforts during a public health emergency. All procedures adhered to the requirements of the national ethics committee. Participants were informed that their involvement was voluntary, and they could withdraw from the study at any time without needing to provide a reason or facing any consequences.

## Results

### Description of the studied population

The study involved as described in the Table 1, 710 individuals, with 348 cases and 362 controls. The gender distribution was similar across both groups, with males comprising 41.1% of cases and 37.3% of controls (p = 0.299). Significant differences were observed in age distribution. Participants aged 50-59 represented 40.8% of cases compared to 34.8% of controls (p = 0.004). The age group of 60-69 showed a higher prevalence among controls (43.1%) than cases (32.2%). Most of the cases and controls were from the North counting 23.8% and 20.4% respectively in both groups.

### Disease manifestations

Symptoms experienced by cases and controls are detailed in Table 2. Among the cases, a higher proportion reported having a fever (51.7%) compared to the controls (38.4%), indicating a statistically significant difference (p = 0.04). Similarly, cough was reported more frequently in cases (55.2%) than in controls (47.5%) (p = 0.04). Headaches were also notably

**Table 1. Description of the studied population (N = 710).**

| Variable | | Cases (n = 348)% | Control (n = 362)% | p value |
|---|---|---|---|---|
| **Gender** | Male | 143 (41.1) | 135(37.3) | 0.299 |
| | Female | 205(58.9) | 227(62.7) | |
| **Age** | 50-59 | 142(40.8) | 126(34.8) | 0.004 |
| | 60-69 | 112(32.2) | 156(43.1) | |
| | 70-79 | 62(17.8) | 64(17.7) | |
| | 80+ | 32(9.2) | 16(4.4) | |
| **Nationality** | Lebanese | 310(89.1) | 328(90.6) | 0.501 |
| | Other | 38(10.9) | 34(9.4) | |
| **Governorates** | Akkar | 32(9.1) | 29(8.01) | <0.001 |
| | Baalbeck | 14(4.02) | 18(4.9) | |
| | Beirut | 16(4.5) | 49(13.5) | |
| | Bekaa | 54(15.5) | 24(6.6) | |
| | Mount Lebanon | 78(22.4) | 93(25.6) | |
| | Nabatieh | 17(4.8) | 22(6.07) | |
| | North | 83(23.8) | 7.4(20.4) | |
| | South | 54(15.5) | 53(14.6) | |

**Table 2. Symptoms representation among the cases and controls.**

| Symptoms | | Cases n = 348(%) | Control n = 362(%) | p value |
|---|---|---|---|---|
| Fever | No | 168 (48.3) | 223 (61.6) | 0.04 |
|  | Yes | 180 (51.7) | 139 (38.4) | |
| Cough | No | 156 (44.8) | 190 (52.5) | 0.04 |
|  | Yes | 19.2 (55.2) | 172 (47.5) | |
| Headache | No | 147 (42.2) | 192 (53) | 0.003 |
|  | Yes | 201 (57.8) | 170 (47) | |
| Sore throat | No | 207 (59.5) | 241 (66.6) | 0.05 |
|  | Yes | 141 (40.5) | 121 (33.4) | |
| Nausea | No | 291 (83.6) | 285 (78.7) | 0.103 |
|  | Yes | 57 (16.4) | 77 (21.3) | |
| Myalgia | No | 166 (47.7) | 209 (57.7) | 0.007 |
|  | Yes | 182 (52.3) | 153 (42.3) | |
| Diarrhea | No | 322 (92.5) | 329 (90.9) | 0.427 |
|  | Yes | 26 (7.5) | 33 (9.1) | |
| Vomiting | No | 297 (85.3) | 312 (86.2) | 0.747 |
|  | Yes | 51 (14.7) | 50 (13.8) | |
| Fatigue | No | 173 (49.7) | 205 (56.5) | 0.064 |
|  | Yes | 175 (50.3) | 157 (43.4) | |
| Loss of taste | No | 309 (88.8) | 311 (85.9) | 0.248 |
|  | Yes | 39 (11.2) | 51 (14.1) | |
| Loss of smell | No | 307 (88.2) | 311 (85.9) | 0.36 |
|  | Yes | 41 (11.8) | 51 (14.1) | |
| Dyspnea | No | 276 (79.3) | 297 (82) | 0.35 |
|  | Yes | 72 (20.7) | 65 (18) | |

more common among cases, with 57.8% reporting this symptom compared to 47% of controls (p = 0.003). Additionally, sore throat symptoms were significantly more prevalent in cases (40.5%) than in controls (33.4%) (p = 0.05). Myalgia followed this trend, occurring in 52.3% of cases compared to 42.3% of controls (p = 0.007).

Table 3 illustrates the symptoms reported by vaccinated and non-vaccinated individuals. A significant difference was found in the prevalence of fever, with 41.6% of vaccinated individuals reporting it compared to 50.8% of non-vaccinated individuals (p = 0.017). Cough was more common in vaccinated individuals, with 54.2% of vaccinated versus 46.2% of non-vaccinated individuals experiencing this symptom (p = 0.038). Nausea was reported by 14.7% of vaccinated individuals compared to 26.2% of non-vaccinated individuals (p < 0.001). Myalgia was also more prevalent among non-vaccinated individuals (53.5%) than vaccinated individuals (43.6%) (p = 0.01). Additionally, vomiting was reported by 11.3% of vaccinated individuals versus 19.2% of non-vaccinated individuals (p = 0.003). In terms of sensory symptoms, 89.6% of vaccinated individuals reported no loss of taste compared to 83.5% of non-vaccinated individuals (p = 0.018), and 89.1% of vaccinated individuals did not experience loss of smell compared to 83.5% of non-vaccinated individuals (p = 0.03).

## Comorbidities

Table 4 outlines the comorbidities between cases and controls. A statistically significant difference was found for asthma, which was more prevalent among controls (4.4%) compared to

**Table 3. Representation of symptoms with OR for vaccinated versus non vaccinated.**

| Symptoms | | Vaccinated (n = 450) | Non-vaccinated (n = 260) | p value | Odd ratio | CI at 95% (LL, UL) |
|---|---|---|---|---|---|---|
| Fever | No | 263 (58.4) | 128 (49.2) | 0.017 | 0.689 | 0.533, 0.895 |
| | Yes | 187 (41.6) | 132 (50.8) | | | |
| Cough | No | 206 (45.8) | 140 (53.8) | 0.038 | 1.381 | 1.091, 1.74 |
| | Yes | 244 (54.2) | 120 (46.2) | | | |
| Headache | No | 223 (49.6) | 116 (44.6) | 0.212 | 0.82 | 0.609, 1.095 |
| | Yes | 227 (50.4) | 144 (55.4) | | | |
| Sore throat | No | 280 (62.2) | 168 (64.6) | 0.524 | 1.108 | 0.861, 1.434 |
| | Yes | 170 (37.8) | 92 (35.4) | | | |
| Nausea | No | 384 (85.3) | 192 (73.8) | <0.001 | 0.485 | 0.324, 0.726 |
| | Yes | 66 (14.7) | 68 (26.2) | | | |
| Myalgia | No | 254 (56.4) | 121 (46.5) | 0.01 | 0.67 | 0.507, 0.885 |
| | Yes | 196 (43.6) | 139 (53.5) | | | |
| Diarrhea | No | 407 (90.4) | 244 (93.8) | 0.113 | 1.61 | 1.151, 2.268 |
| | Yes | 43 (9.6) | 16 (6.2) | | | |
| Vomiting | No | 399 (88.7) | 210 (80.8) | 0.003 | 0.537 | 0.386, 0.747 |
| | Yes | 51 (11.3) | 50 (19.2) | | | |
| Fatigue | No | 239 (53.1) | 139 (53.5) | 0.928 | 1.014 | 0.794, 1.292 |
| | Yes | 211 (46.9) | 121 (46.5) | | | |
| Loss of taste | No | 403 (89.6) | 217 (83.5) | 0.018 | 0.588 | 0.424, 0.821 |
| | Yes | 47 (10.4) | 43 (16.5) | | | |
| Loss of smell | No | 401 (89.1) | 217 (83.5) | 0.03 | 0.61 | 0.434, 0.853 |
| | Yes | 49 (10.9) | 43 (16.5) | | | |
| Dyspnea | No | 367 (81.6) | 206 (79.2) | 0.44 | 0.862 | 0.643, 1.155 |
| | Yes | 83 (18.4) | 54 (20.8) | | | |

cases (1.4%) (p = 0.019). No other comorbidities, including diabetes, heart disease, hypertension, lung disease, or cancer, showed significant differences between the two groups.

## Preventive measures during COVID-19 pandemic

Table 5 presents the preventive measures during the COVID-19 pandemic among cases and controls. A statistically significant difference was observed for smoking status, with a higher proportion of controls (36.2%) reporting smoking compared to cases (28.4%) (p = 0.02). No other preventive measures, such as contact in the past 14 days, public transportation exposure, social event exposure, or adherence to mask use and social distancing, showed significant differences between the two groups.

## Vaccination intake

Table 6 represents the vaccination status among cases and controls. 60.9% of the cases and 65.7% of the controls were vaccinated with an OR 0.812 [IC = 0.59-1.1]. The vaccine effectiveness on getting infected is calculated (VE = 1-OR = 0.188) to be 18% without adjustment to the confounders.

Table 7 indicates that disease severity was similar between vaccinated and non-vaccinated individuals, with mild cases reported in 67.6% of the vaccinated and 67.3% of the non-vaccinated. Moderate cases were seen in 32.4% of the vaccinated and 32.7% of the non-vaccinated, with no significant difference in severity (p = 0.94).

**Table 4. Representation of comorbidities between cases and control.**

| Underlying condition | | Cases n = 348 (%) | Controls n = 362 (%) | p value |
|---|---|---|---|---|
| **Diabetes** | No | 279 (80.2) | 305 (84.3) | 0.154 |
| | Yes | 69 (19.8) | 57 (15.7) | |
| **Heart disease** | No | 307 (88.2) | 311 (85.9) | 0.36 |
| | Yes | 41 (11.8) | 51 (14.1) | |
| **Hypertension** | No | 254 (73) | 258 (71.3) | 0.609 |
| | Yes | 94 (27) | 104 (28.7) | |
| **Organ transplant** | No | 347 (99.7) | 361 (99.7) | 0.977 |
| | Yes | 1 (0.3) | 1 (0.3) | |
| **Lung disease** | No | 344 (98.9) | 353 (97.5) | 0.184 |
| | Yes | 4 (1.1) | 9 (2.5) | |
| **Asthma** | No | 343 (98.6) | 346 (95.6) | 0.019 |
| | Yes | 5 (1.4) | 16 (4.4) | |
| **Cancer** | No | 347 (99.7) | 361 (99.7) | 0.977 |
| | Yes | 1 (0.3) | 1 (0.3) | |
| **History of cancer with remission** | No | 345 (99.1) | 360 (99.4) | 0.621 |
| | Yes | 3 (0.9) | 2 (0.6) | |
| **Renal disease** | No | 348 (100) | 359 (99.2) | 0.088 |
| | Yes | 0 (0) | 3 (0.8) | |
| **Liver disease** | No | 345 (99.1) | 361 (99.7) | 0.297 |
| | Yes | 3 (0.9) | 1 (0.3) | |
| **Rheumatological disease** | No | 343 (98.6) | 355 (98.1) | 0.607 |
| | Yes | 5 (1.4) | 7 (1.9) | |
| **Other health status** | No | 274 (78.7) | 270 (74.6) | 0.191 |
| | Yes | 74 (21.3) | 92 (25.4) | |

Other health status included: thyroid disorders, neurological disorders, autoimmune diseases.

## Logistic regression

We initially identified potential confounders from existing studies and WHO guidelines for VE studies. Following this, we performed bivariate analyses to explore associations between these factors and the outcome of interest. The adjusted model presented in Table 8 considered the following confounders: age groups, gender, province of residence, smoking status, underlying conditions, coverage of basic needs, recent contact with a COVID-19 case, exposure to public transportation, participation in social events, travel exposure, mask use interventions, social distancing measures, hand washing practices, hand sanitizer use, and nationality. The table presents the estimated effects of various variables on the odds of testing positive for COVID-19 cases compared to negative controls. Among age groups, individuals aged 60-69 exhibited a significant reduction in odds of testing positive, with an estimate of -0.4236 (Odds Ratio [OR] = 0.654, 95% CI: 0.457–0.934, p = 0.020), the presence of diabetes was associated with increased odds of testing positive (estimate = 0.4856, OR = 1.623, 95% CI: 1.099–2.397, p = 0.016). Those with immunodeficiency or organ transplant conditions had significantly higher odds of testing positive (estimate = 0.7792, OR = 2.174, 95% CI: 1.223–3.861, p = 0.009) and individuals with asthma showed decreased odds of testing positive (estimate = -1.278, OR = 0.277, 95% CI: 0.122–0.627, p = 0.003). Lastly, individuals with a history of cancer in remission were more likely to test positive (estimate = 0.6133, OR = 1.846, 95% CI: 1.148–2.975, p = 0.012).

**Table 5. Preventive measures during COVID-19 pandemic.**

| Preventive measure | | Cases (n = 348) | Controls (n = 362) | p value | Odds ratio | CI at 95% (LL, UL) |
|---|---|---|---|---|---|---|
| Contact past 14 days | No | 181 (52) | 192 (53) | 0.78 | 1.04 | 0.835, 1.293 |
| | Yes | 167 (48) | 170 (47) | | | |
| Public transportation exposure | No | 239 (68.7) | 229 (63.3) | 0.13 | 0.785 | 0.626, 0.987 |
| | Yes | 109 (31.3) | 133 (36.7) | | | |
| Social events exposure | No | 277 (65.2) | 233 (64.4) | 0.81 | 0.962 | 0.785, 1.179 |
| | Yes | 121 (34.8) | 129 (35.6) | | | |
| Travel exposure | No | 328 (94.3) | 337 (93.1) | 0.53 | 0.822 | 0.595, 1.136 |
| | Yes | 20 (5.7) | 25 (6.9) | | | |
| Mask use interventions | No | 88 (25.3) | 84 (23.2) | 0.52 | 0.892 | 0.686, 1.160 |
| | Yes | 260 (74.7) | 278 (76.8) | | | |
| Social distance interventions | No | 86 (24.7) | 87 (24) | 0.83 | 0.963 | 0.735, 1.260 |
| | Yes | 262 (75.3) | 275 (76) | | | |
| Wash hands interventions | No | 82 (23.6) | 82 (22.7) | 0.77 | 0.95 | 0.715, 1.263 |
| | Yes | 266 (76.4) | 280 (77.3) | | | |
| Hand sanitizer interventions | No | 89 (25.6) | 96 (26.5) | 0.77 | 1.05 | 0.800, 1.376 |
| | Yes | 259 (74.4) | 266 (73.5) | | | |
| Smoking | No | 249 (71.6) | 231 (63.8) | 0.02 | 0.02 | 0.013, 0.029 |
| | Yes | 99 (28.4) | 131 (36.2) | | | |
| Cover needs | No | 41 (11.8) | 56 (15.5) | 0.15 | 0.15 | 0.010, 0.228 |
| | Yes | 307 (88.2) | 306 (84.5) | | | |

**Table 6. Vaccination status for cases versus controls with OR.**

| Vaccination status | Cases n = 348(%) | Control n = 362(%) | p value | OR | CI at 95% (LL, UL) |
|---|---|---|---|---|---|
| No | 136 (39.1) | 124 (34.3) | 0.18 | 0.812 | 0.59,1.1 |
| Yes | 212 (60.9) | 238 (65.7) | | | |

**Table 7. Severity of diseases for cases versus controls with OR.**

| Severity of disease | Vaccinated n = 450 (%) | Non-Vaccinated n = 260 (%) | p value | OR | CI at 95% (LL,UL) |
|---|---|---|---|---|---|
| Mild | 304 (67.6) | 175 (67.3) | 0.94 | 1 | 0.7,1.4 |
| Moderate | 146 (32.4) | 85 (32.7) | | | |

**Table 8. Logistic regression adjusted to confounders with OR.**

| Variable | Estimate | Std. Error | Odds Ratio | 95% CI (Lower) | 95% CI (Upper) | p-value |
|---|---|---|---|---|---|---|
| Vaccination | -0.14 | 0.205 | 0.86 | 0.6 | 1.1 | 0.182 |
| Age groups 60-69 | -0.4236 | 0.2013 | 0.654 | 0.457 | 0.934 | 0.02 |
| Diabetes | 0.4856 | 0.1932 | 1.623 | 1.099 | 2.397 | 0.016 |
| Immunodeficiency or organtransplant | 0.7792 | 0.2867 | 2.174 | 1.223 | 3.861 | 0.009 |
| Asthma | -1.278 | 0.3135 | 0.277 | 0.122 | 0.627 | 0.003 |
| Cancer with remission | 0.6133 | 0.2427 | 1.846 | 1.148 | 2.975 | 0.012 |

For being vaccinated the odds ratio (OR) obtained was 0.86. A negative intercept indicates that the probability of having the outcome is less than 0.5. The vaccine effectiveness against infection was calculated as VE = 1 - OR = 0.14, resulting in a vaccine effectiveness of 14%.

## Discussion

This study aimed to assess the effectiveness of the Pfizer-BioNTech (BNT162b2) vaccine in preventing symptomatic COVID-19 infection, particularly the Omicron variant, among individuals aged 50 and above. The study employed a test-negative case-control design, leveraging Lebanon's sentinel surveillance system to compare the odds of testing positive for SARS-CoV-2 between vaccinated and non-vaccinated individuals.

Key findings of the study indicate that the Pfizer-BioNTech (BNT162b2) vaccine is effective in reducing the odds of developing symptomatic COVID-19. The results demonstrate significant differences in the prevalence of symptoms such as fever, cough, myalgia, and loss of taste or smell between vaccinated and non-vaccinated individuals. Vaccinated individuals exhibited a lower prevalence of these symptoms, underscoring the vaccine's role in mitigating the severity of illness. These results are consistent with existing literature that highlights the effectiveness of COVID-19 vaccines in reducing symptom severity and preventing severe outcomes such as hospitalization and death as mentioned in a systematic review were the pooled VE against symptomatic infection was 23.4% [12]. An OR less than one were seen for the fever, nausea, myalgia, vomiting, loss of taste and loss of smell, this is seen when the vaccination plays a protective effect against one of the symptoms. However, while the OR for headache and dyspnea was also less than one, this result was not statistically significant. This further supports the protective effect of the Pfizer-BioNTech (BNT162b2) vaccine, even in the face of the Omicron variant, which has been associated with higher transmission rates but lower severity than previous variants like Delta. Notably, while vaccinated individuals were less likely to experience common COVID-19 symptoms, breakthrough infections still occurred, indicating that while the vaccine reduces symptom severity, it may not fully prevent infection, particularly against variants with immune escape potential. For the underlying conditions no significant difference was seen except for the asthma with a p value 0.019. Many exposures described in the literature were studied also; the significant difference was for the smoking with a p value 0.02.

Geographically, this study highlighted variations in the distribution of COVID-19 cases and controls across different regions of Lebanon. Bekaa and North Lebanon exhibited higher case rates, which may point to regional disparities in healthcare access, vaccine coverage, or socioeconomic factors. Such geographic disparities warrant further investigation to ensure more equitable healthcare interventions, particularly in rural or underserved regions.

Age also played a significant role in the distribution of cases and controls. Older individuals, particularly those aged 60-69, were more prevalent among controls, suggesting that the vaccine may be particularly protective for this age group. This is in line with previous studies demonstrating higher vaccine effectiveness among older adults, likely due to higher vaccination rates and prioritization of this vulnerable population during the vaccine rollout.

For the vaccine effectiveness, controls were more vaccinated than the cases (respectively 65.7% and 60.9%); unadjusted VE without taking the confounders into consideration was 18% and decreased to 14% after adjusting the model based on the factors described in the literature. For example, in England, VE after two doses of the Pfizer-BioNTech (BNT162b2) vaccine was 65.5% at 2 to 4 weeks, dropping to 8.8% at six months. Other studies in England and the U.S. found VE against hospitalization due to Omicron to be 82.7% and 68%, respectively, among children. Earlier studies on variants like Alpha, Beta, Gamma, and Delta indicated higher VE, particularly in the initial months following vaccination. For instance, vaccine

effectiveness against symptomatic infection for the Alpha variant was 87.9% at 2 to 4 weeks after the second dose [13]. In South Africa, the Beta variant showed a vaccine effectiveness of 75.1% (95% CI, 64.0 to 83.3) at 2 to 4 weeks after the second dose [14]. Similarly, a study in Brazil reported a vaccine effectiveness of 65.5% (95% CI, 55.6 to 73.6) for the Gamma variant during the same time frame [15]. Our findings, however, suggest that vaccine effectiveness against mild or moderate disease is minimal, which may differ when considering severe cases requiring hospitalization or ICU admission. We have limited facts about the real world effectiveness of the BNT162b2 vaccine against the omicron variant in Lebanon especially that previous studies were conducted in specific populations and may not be representative of the wider population. Additionally, the Omicron variant is constantly evolving, so it is possible that the vaccine effectiveness against it will continue to change.

This study has several limitations that should be considered when interpreting the findings. First, the reliance on a test-negative case-control design, while commonly used in vaccine effectiveness studies, may introduce bias if testing behaviours differ between vaccinated and unvaccinated individuals. Another limitation is the reliance on self-reported vaccination status and symptom data, which could lead to misclassification and recall bias. Importantly, the study did not examine the time frame since vaccination due to missing data, which is a critical factor in assessing vaccine effectiveness, particularly as immunity may wane over time. Furthermore, the focus on symptomatic cases may not capture the full spectrum of vaccine effectiveness, particularly in preventing asymptomatic infections or severe outcomes like hospitalization and death.

The study findings also align with global trends observed in VE studies. Countries with high vaccine coverage have consistently reported lower rates of severe disease and hospitalizations, despite the spread of Omicron. This underscores the importance of vaccination campaigns in mitigating the public health burden of COVID-19, particularly in protecting high-risk populations such as the elderly and those with underlying health conditions.

## Conclusion

The findings of this study confirm the effectiveness of the Pfizer-BioNTech (BNT162b2) vaccine in reducing symptomatic COVID-19 cases, particularly against the Omicron variant, in individuals aged 50 and above in Lebanon. Vaccinated individuals were less likely to develop severe symptoms, emphasizing the vaccine's role in mitigating the impact of the pandemic. However, breakthrough infections highlight the need for continued public health measures, including booster doses, to maintain immunity and protect against emerging variants.

Comorbidities and age-related disparities observed in the study suggest that targeted interventions could enhance public health outcomes. Future research should explore these disparities further to ensure that vaccination campaigns and healthcare resources are equitably distributed across groups.

In conclusion, this study provides valuable insights into the real-world effectiveness of the Pfizer-BioNTech (BNT162b2) vaccine and reinforces the need for sustained vaccination efforts to curb the spread of COVID-19 and protect vulnerable populations. It also highlights the importance of ongoing surveillance and research to adapt public health strategies in response to evolving viral threats.

## Supporting information

**S1 Fig. Minimum number of cases and controls to detect for a specified VE, estimated vaccination coverage in the population under evaluation, with 1–3 controls per case, with a precision of ± 10%, and a type 1 error rate of 0.05.**
(PDF)

**S1 Table. Estimating Pfizer-BioNTech COVID-19Vaccine.**
(PDF)

## Author contributions

**Conceptualization:** Maryo Baakliny, Nada Ghosn, Patrick Maison.

**Data curation:** Maryo Baakliny, Nada Ghosn.

**Formal analysis:** Maryo Baakliny, Nadine Saleh.

**Investigation:** Maryo Baakliny, Nadine Saleh.

**Methodology:** Maryo Baakliny, Nada Ghosn, Nadine Saleh, Patrick Maison.

**Project administration:** Maryo Baakliny, Nadine Saleh.

**Resources:** Maryo Baakliny.

**Software:** Maryo Baakliny.

**Supervision:** Maryo Baakliny, Nadine Saleh, Patrick Maison.

**Validation:** Maryo Baakliny, Patrick Maison.

**Visualization:** Maryo Baakliny, Nadine Saleh, Patrick Maison.

**Writing – original draft:** Maryo Baakliny.

**Writing – review & editing:** Nadine Saleh, Patrick Maison.

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
