## [Decision Letter · Decision Letter 0]

16 Sep 2024

PONE-D-24-09717Effectiveness of the Pfizer-BioNTech Vaccine against the Omicron Variant of SARS-CoV-2 among Adults Aged 50 and Above: A Case-Control Study in Lebanon, June 2022PLOS ONE

Dear Dr. Baakliny,

Thank you for submitting your manuscript to PLOS ONE. After careful consideration, we feel that it has merit but does not fully meet PLOS ONE’s publication criteria as it currently stands. Therefore, we invite you to submit a revised version of the manuscript that addresses the points raised during the review process.

We look forward to receiving your revised manuscript.

Kind regards,

Marwan Osman

Academic Editor

PLOS ONE

Journal requirements: 1. When submitting your revision, we need you to address these additional requirements. Please ensure that your manuscript meets PLOS ONE's style requirements, including those for file naming. The PLOS ONE style templates can be found at https://journals.plos.org/plosone/s/file?id=wjVg/PLOSOne_formatting_sample_main_body.pdf and https://journals.plos.org/plosone/s/file?id=ba62/PLOSOne_formatting_sample_title_authors_affiliations.pdf. 2. Please ensure you have stated the date of data collection in the Methods section of your manuscript text to fully comply with the PLOS ONE policy on reporting research involving human participants. This information is currently provided only in the Human Participants Research Checklist, which will not be published with your manuscript files.  3. When completing the data availability statement of the submission form, you indicated that you will make your data available on acceptance. We strongly recommend all authors decide on a data sharing plan before acceptance, as the process can be lengthy and hold up publication timelines. Please note that, though access restrictions are acceptable now, your entire data will need to be made freely accessible if your manuscript is accepted for publication. This policy applies to all data except where public deposition would breach compliance with the protocol approved by your research ethics board. If you are unable to adhere to our open data policy, please kindly revise your statement to explain your reasoning and we will seek the editor's input on an exemption. Please be assured that, once you have provided your new statement, the assessment of your exemption will not hold up the peer review process. 4. Please include a copy of Table 9 which you refer to in your text on page 11. 5. Please include captions for your Supporting Information files at the end of your manuscript, and update any in-text citations to match accordingly. Please see our Supporting Information guidelines for more information: http://journals.plos.org/plosone/s/supporting-information. 

Reviewers' comments:

Reviewer's Responses to Questions

**Comments to the Author**

1. Is the manuscript technically sound, and do the data support the conclusions?

Reviewer #1: Yes

Reviewer #2: Partly

2. Has the statistical analysis been performed appropriately and rigorously? 

Reviewer #1: Yes

Reviewer #2: I Don't Know

3. Have the authors made all data underlying the findings in their manuscript fully available?

Reviewer #1: Yes

Reviewer #2: Yes

4. Is the manuscript presented in an intelligible fashion and written in standard English?

Reviewer #1: Yes

Reviewer #2: No

5. Review Comments to the Author

Reviewer #1: Comments to the author

Thank you for the opportunity to review the manuscript “Effectiveness of the Pfizer-BioNTech Vaccine against the Omicron Variant of SARSCoV-2 among Adults Aged 50 and Above: A Case-Control Study in Lebanon, June 2022”.

The study assessed vaccine effectiveness and adjusted the model based on confounders, providing a nuanced understanding of vaccine effectiveness in real-world settings. This methodology can be adopted for various vaccines in future pandemics.

The manuscript is technically sound and supports its conclusions effectively. The statistical analysis is performed rigorously, with appropriate methods and sample sizes ensuring reliable results. The manuscript is presented clearly and written in standard English, making it easily understandable. The data presented is well-aligned with the conclusions drawn. Overall, the study is well-executed and adheres to the required standards of clarity and scientific rigor. However, minor revisions are needed to improve the clarity and structure of the article.

Comment 1: In the abstract, there are redundant ideas present in these two paragraphs. It is preferable to consolidate this idea into a single, clear statement to avoid duplication.

“Factors such as age, province of residence, and underlying conditions significantly influenced VE. The study underscores the need for on-going monitoring and potential booster doses to enhance protection, especially against evolving variants like Omicron.”

“Factors such as age, province of residence, and underlying conditions exert significant influence on VE, highlighting the need for tailored vaccination strategies. Booster doses may play a crucial role in enhancing VE and mitigating the impact of evolving strains like Omicron.”

Comment 2: There seems to be an error in the percentage calculations in the following paragraph in the introduction section. Please review and update the percentages accordingly.

“Whereas in Lebanon as of 17 February 2022 5,779,829 vaccine administered in total (7). The Pfizer, BioNTech vaccine was the most administered with 4,253,218 doses (82,9%) followed by AstraZeneca 717,304 (13.9%); added together they make 96.8 % of the brands administered.”

Comment 3: “Introduction: This study aims to highlight the effectiveness of the Pfizer-BioNTech vaccine brand against the Omicron variant of SARS-CoV-2, focusing on individuals aged 50 and above presenting with influenza-like illness at sentinel sites.”

Could you please explain why you chose to study this specific population? If there is a particular reason, it would be more informative to include this rationale in the methodology section of the article.

Comment 4: Please include relevant dates in the methodology section, specifically the periods of patient recruitment.

Comment 5: The methodology is clear and well written. However, I recommend creating a specific section for data analysis instead of merging it with study design or data collection sections. For example:

“The outcome of interest for the primary analysis will be the detection of SARS-CoV-2 in patients eligible for vaccination and presented to ILI/CLI sites with symptoms fitting the case definition.” This information was stated in the study design section. However, this should be mentioned in the data analysis section.

“Data analysis was done using R version 4.0.2 and R Studio. The measure of association is an odds ratio (OR). This can be measured by logistic regression taking into considerations all the confounding factors cited in the literature.” This information was stated in the data collection section. However, this should be mentioned in the data analysis section.

In addition, it would be beneficial to provide more details on how quantitative variables were handled in the analyses and describe all statistical methods used.

Comment 6: How were the confounders chosen? Did you base your selection solely on the literature, or did you also consider the results of the bivariate analysis?

Comment 7: The result section is well-structured. However, there is a replication of results between the text and the tables; for example, the disease manifestations paragraph repeats all the results of table 2. To avoid duplication, it is recommended to report only the most prevalent manifestations or the significant results in the text. The remaining variables can be presented in the table.

Comment 8: In the result section, specifically the “Table 2: Logistic regression adjusted to confounders with OR”, it is recommended to add the confidence interval and p-value of the OR.

In addition, in this same table, it would be beneficial to add the results of the confounding variables considered in the logistic regression model, and not to present only the vaccination status variable. (you can present at least the variables with significant results).

Comment 9: In the discussion section, you mentioned that: “An OR less than one were seen for the fever, headache, nausea, myalgia, vomiting, loss of taste, loss of smell this is seen when the vaccination plays a protective effect against one of the symptoms.” However, according to the table 3, the P-value for “headache” is 0.212, which is a not statistically significant. This needs to be adjusted in the text.

Comment 10: The conclusion section is well-written, summarizing the main results and outlining the implications for future research. However, I suggested moving the first paragraph of the conclusion (“Knowing that we have limited facts  for the Gamma variant during the same time frame (17)”) to the discussion section. This paragraph compares the study results with other studies conducted in the UK and South Africa, making it more appropriate for the discussion section.

Comment 11: Please review the numbering of the tables.

Reviewer #2: The study titled “Effectiveness of the Pfizer-BioNTech Vaccine against the Omicron Variant of

SARS-CoV-2 among Adults Aged 50 and Above: A Case-Control Study in Lebanon,

June 2022” focuses on an important topic however various issues are associated with the manuscript. The below recommendations should be paid attention to in order to ensure proper presentation of the paper and utilization/interpretation of results.

-Please ensure to state what acronym stand for when the acronym is used for the first time (eg. SARI in study design instead of study population).

-Also please keep consistent towards the terminologies used and avoid typos (SARS-CoV-2 vs SARSCoV-2 in study design section).

-Ethical consideration section should have been in past tense (why were “will” and “should” included in this part?)

- The manuscript needs to go through an overhaul in terms of organization. It is unclear what the table containing the chronic diseases right underneath table 1 is? Also, table formats should be consistent.

- Please keep spacing and spacing before/after punctuation consistent.

- The manuscript should be read very carefully to correct typos and grammatical issues. Many typos and punctuation errors exist.

- A more elaborate discussion / conclusion is required to ensure that results are thoroughly discussed and elaborated on.

-The authors should check to see if there is any correlation between the timing of vaccination and the symptoms observed. For instance, previous work (Barin et al., Lancet Microbe, 2022 and others) demonstrated a significant reduction in antibody titers over time so it is crucial to observe if the symptom severity is reduced in recently vaccinated individuals. Please carry out correlation studies in the light of different vaccination times and symptoms experienced.

-Why weren’t the results for thyroid disorders, neurological disorders, autoimmune

Diseases included in the report? This could provide interesting and novel information.

6. PLOS authors have the option to publish the peer review history of their article (what does this mean? ). If published, this will include your full peer review and any attached files.

**Do you want your identity to be public for this peer review?** For information about this choice, including consent withdrawal, please see our Privacy Policy .

Reviewer #1: No

Reviewer #2: No

---

## [Author Response · Author response to Decision Letter 1]

6 Nov 2024

We sincerely appreciate the reviewers' thoughtful feedback and have made revisions accordingly to enhance the clarity and depth of our manuscript. Below is a detailed table that outlines each of the reviewers' comments, along with our corresponding responses and the actions taken to address them.

Response to Reviewer One: Thank you for your constructive feedback, which helped improve the clarity and quality of our manuscript.

Comment 1: In the abstract, there are redundant ideas present in these two paragraphs. It is preferable to consolidate this idea into a single, clear statement to avoid duplication.

“Factors such as age, province of residence, and underlying conditions significantly influenced VE. The study underscores the need for on-going monitoring and potential booster doses to enhance protection, especially against evolving variants like Omicron.”

“Factors such as age, province of residence, and underlying conditions exert significant influence on VE, highlighting the need for tailored vaccination strategies. Booster doses may play a crucial role in enhancing VE and mitigating the impact of evolving strains like Omicron.” We agree that the two paragraphs in the abstract contained redundant ideas. We have revised and consolidated them into a single, clearer statement as follows:

Factors such as age, province of residence, and underlying conditions significantly influenced VE. The study underscores the need for an on-going monitoring and potential booster doses to enhance protection, especially against evolving variants like Omicron and emphasizes the importance of proactive measures in safeguarding public health amidst evolving viral threats.

Comment 2: There seems to be an error in the percentage calculations in the following paragraph in the introduction section. Please review and update the percentages accordingly.

“Whereas in Lebanon as of 17 February 2022 5,779,829 vaccine administered in total (7). The Pfizer, BioNTech vaccine was the most administered with 4,253,218 doses (82,9%) followed by AstraZeneca 717,304 (13.9%); added together they make 96.8 % of the brands administered.” We have reviewed the numbers and corrected the percentages accordingly. The revised paragraph now reads:

Whereas in Lebanon as of 17 February 2022 5,779,829 vaccine had been administered in total by the same time[7] .The Pfizer, BioNTech vaccine was the most administered with 4,253,218 doses (73.58 %) followed by AstraZeneca 717,304 (12.41%); added together they make 85.9% of the total brands administered.

Comment 3: “Introduction: This study aims to highlight the effectiveness of the Pfizer-BioNTech vaccine brand against the Omicron variant of SARS-CoV-2, focusing on individuals aged 50 and above presenting with influenza-like illness at sentinel sites.”

Could you please explain why you chose to study this specific population? If there is a particular reason, it would be more informative to include this rationale in the methodology section of the article. We chose to focus on individuals aged 50 and above in this study because in Lebanon, the Pfizer-BioNTech vaccine was primarily administered to healthcare workers and individuals in this age group due to their higher risk of severe outcomes from COVID-19. Clarification was added in the methodology section.

Comment 4: Please include relevant dates in the methodology section, specifically the periods of patient recruitment. We have updated the methodology section to include the relevant dates for patient recruitment. The revised sentence now reads:

Data collection was conducted from February till the end of April.

Comment 5: The methodology is clear and well written. However, I recommend creating a specific section for data analysis instead of merging it with study design or data collection sections. For example:

“The outcome of interest for the primary analysis will be the detection of SARS-CoV-2 in patients eligible for vaccination and presented to ILI/CLI sites with symptoms fitting the case definition.” This information was stated in the study design section. However, this should be mentioned in the data analysis section.

“Data analysis was done using R version 4.0.2 and R Studio. The measure of association is an odds ratio (OR). This can be measured by logistic regression taking into considerations all the confounding factors cited in the literature.” This information was stated in the data collection section. However, this should be mentioned in the data analysis section.

In addition, it would be beneficial to provide more details on how quantitative variables were handled in the analyses and describe all statistical methods used. A new section was added as suggested :

Data analysis:

The outcome of interest for the primary analysis was the detection of SARS-CoV-2 in patients eligible for vaccination and presented to ILI/CLI sites with symptoms fitting the case definition. Data analysis was done using R version 4.0.2 and R Studio. Descriptive statistics were calculated to summarize the demographic and clinical characteristics of the participants, frequencies and numbers were reported for categorical variables and comparisons of percentages between groups were made using Pearson's Chi-square tests for categorical variables to assess the significance of observed differences. The measure of association was the odds ratio (OR). This was can be measured by the logistic regression taking into considerations all the confounding factors cited in the literature.

Comment 6: How were the confounders chosen? Did you base your selection solely on the literature, or did you also consider the results of the bivariate analysis? Explanation was added to the methods section:

The selection of confounders was based on the guideline by WHO for VE studies. Following this, we performed bivariate analyses to explore associations between these factors and the outcome of interest and all variables with p value less than 0.2 were added. This dual approach allowed us to ensure that our confounder selection was comprehensive and grounded in both theoretical and empirical evidence. The chosen confounders included age groups, gender, province of residence, smoking status, underlying conditions, coverage of basic needs, recent contact with a COVID-19 case, exposure to public transportation, social events, travel, mask use, social distancing, handwashing, hand sanitizer use, and nationality.

Comment 7: The result section is well-structured. However, there is a replication of results between the text and the tables; for example, the disease manifestations paragraph repeats all the results of table 2. To avoid duplication, it is recommended to report only the most prevalent manifestations or the significant results in the text. The remaining variables can be presented in the table. To enhance clarity and reduce redundancy, we revised the results section and adjusted accordingly.

Comment 8: In the result section, specifically the “Table 2: Logistic regression adjusted to confounders with OR”, it is recommended to add the confidence interval and p-value of the OR.

In addition, in this same table, it would be beneficial to add the results of the confounding variables considered in the logistic regression model, and not to present only the vaccination status variable. (you can present at least the variables with significant results). We have revised “Table 2” and adjusted the title to “table 8”: Logistic regression adjusted to confounders with OR” to include the confidence intervals (CIs) and p-values for the odds ratios (OR), as requested. Additionally, we have expanded the table to include the results for the confounding variables considered in the logistic regression model.

Comment 9: In the discussion section, you mentioned that: “An OR less than one were seen for the fever, headache, nausea, myalgia, vomiting, loss of taste, loss of smell this is seen when the vaccination plays a protective effect against one of the symptoms.” However, according to the table 3, the P-value for “headache” is 0.212, which is a not statistically significant. This needs to be adjusted in the text. Adjusted upon your comment to be: However, while the OR for headache and dyspnea was also less than one, this result was not statistically significant.

Comment 10: The conclusion section is well-written, summarizing the main results and outlining the implications for future research. However, I suggested moving the first paragraph of the conclusion (“Knowing that we have limited facts  for the Gamma variant during the same time frame (17)”) to the discussion section. This paragraph compares the study results with other studies conducted in the UK and South Africa, making it more appropriate for the discussion section. We agree that the first paragraph of the conclusion, which compares the study results with other studies conducted in the UK and South Africa, is more suited to the discussion section. We have now moved the paragraph to the discussion section.

Comment 11: Please review the numbering of the tables. Reviewed and adjusted.

Response to Reviewer Two: thank you for your detailed review and insightful suggestions. Your input has significantly contributed to refining our work.

Comment 1: Please ensure to state what acronym stand for when the acronym is used for the first time (eg. SARI in study design instead of study population). It was adjusted accordingly and we have also reviewed the document to confirm that other acronyms are similarly defined upon their first use to enhance clarity for readers.

Comment 2: Also please keep consistent towards the terminologies used and avoid typos (SARS-CoV-2 vs SARSCoV-2 in study design section). We have carefully reviewed the manuscript to ensure that the terminology is consistent throughout the document.

Comment 3: Ethical consideration section should have been in past tense (why were “will” and “should” included in this part?) We have revised the section accordingly.

Comment 4: The manuscript needs to go through an overhaul in terms of organization. It is unclear what the table containing the chronic diseases right underneath table 1 is? Also, table formats should be consistent. Thank you for your feedback. We recognize the importance of a clear and consistent structure throughout the manuscript. The table containing chronic diseases beneath Table 1 was unintentionally misplaced and was relocated to the appropriate section for clarity. Additionally, we ensured that all table formats are consistent.

Comment 5: Please keep spacing and spacing before/after punctuation consistent. Thank you for highlighting this. We have thoroughly reviewed the entire manuscript to ensure consistency in spacing.

Comment 6: The manuscript should be read very carefully to correct typos and grammatical issues. Many typos and punctuation errors exist. The manuscript has undergone a comprehensive review to address and correct all typos, grammatical issues, and punctuation errors.

Comment 7: A more elaborate discussion / conclusion is required to ensure that results are thoroughly discussed and elaborated on. We have expanded both the discussion and conclusion sections to provide a more comprehensive analysis of our results.

Comment 8: The authors should check to see if there is any correlation between the timing of vaccination and the symptoms observed. For instance, previous work (Barin et al., Lancet Microbe, 2022 and others) demonstrated a significant reduction in antibody titers over time so it is crucial to observe if the symptom severity is reduced in recently vaccinated individuals. Please carry out correlation studies in the light of different vaccination times and symptoms experienced. Thank you for your valuable suggestion regarding the correlation between the timing of vaccination and the symptoms observed.

However, due to the constraints of our current dataset, we were unable to perform a thorough correlation analysis between vaccination timing and the symptoms experienced by participants. We emphasized this limitation in the revised manuscript and suggest it as a direction for future studies.

Comment 9: Why weren’t the results for thyroid disorders, neurological disorders and autoimmune diseases included in the report? This could provide interesting and novel information. The primary reason these results were not included in the report is due to limitations in our sample size and the statistical power needed to analyze these specific disorders. During the analysis, we found that the prevalence of these conditions within our study population was low and were cited as others in free text, which limited the ability to draw meaningful conclusions or to perform robust statistical analyses.

---

## [Decision Letter · Decision Letter 1]

17 Dec 2024

PONE-D-24-09717R1Effectiveness of the Pfizer-BioNTech Vaccine against the Omicron Variant of SARS-CoV-2 among Adults Aged 50 and Above: A Case-Control Study in Lebanon, June 2022PLOS ONE

Dear Dr. Baakliny,

Thank you for submitting your manuscript to PLOS ONE. After careful consideration, we feel that it has merit but does not fully meet PLOS ONE’s publication criteria as it currently stands. Therefore, we invite you to submit a revised version of the manuscript that addresses the points raised during the review process.

We look forward to receiving your revised manuscript.

Kind regards,

Marwan Osman, PhD, MPH

Academic Editor

PLOS ONE

Journal Requirements:

Reviewers' comments:

Reviewer's Responses to Questions

**Comments to the Author**

1. If the authors have adequately addressed your comments raised in a previous round of review and you feel that this manuscript is now acceptable for publication, you may indicate that here to bypass the “Comments to the Author” section, enter your conflict of interest statement in the “Confidential to Editor” section, and submit your "Accept" recommendation.

Reviewer #1: All comments have been addressed

Reviewer #3: All comments have been addressed

2. Is the manuscript technically sound, and do the data support the conclusions?

Reviewer #1: Yes

Reviewer #3: Yes

3. Has the statistical analysis been performed appropriately and rigorously? 

Reviewer #1: Yes

Reviewer #3: I Don't Know

4. Have the authors made all data underlying the findings in their manuscript fully available?

Reviewer #1: Yes

Reviewer #3: Yes

5. Is the manuscript presented in an intelligible fashion and written in standard English?

Reviewer #1: Yes

Reviewer #3: Yes

6. Review Comments to the Author

Reviewer #1: All the comments have been addressed by the author which make the manuscript more organized and clearer. No additional comments are requested.

Reviewer #3: My previous comments have been sufficiently addressed. Some things to add would be to state which version of Pfizer Biontech vaccine was implemented and considered during study time as many different updated vaccines emerged. Some of those were more effective against omicron than others. Similarly, please clarify what is meant by “fully vaccinated” in the “Definition of vaccination status:” section highlighting which vaccine version was considered. The originally implemented ones? If so please clarify accordingly.

I agree with the statement in the discussion which states “ Key findings of the study indicate that the Pfizer-BioNTech vaccine is effective in reducing the odds of developing symptomatic COVID-19.” This should have been highlighted more and emphasized in the abstract as it is an important finding. The abstract reads almost as vaccination not having an enough positive impact where the authors' key findings clearly indicates otherwise. Please update accordingly.

7. PLOS authors have the option to publish the peer review history of their article (what does this mean? ). If published, this will include your full peer review and any attached files.

**Do you want your identity to be public for this peer review?** For information about this choice, including consent withdrawal, please see our Privacy Policy .

Reviewer #1: No

Reviewer #3: No

---

## [Author Response · Author response to Decision Letter 2]

7 Jan 2025

Greetings,

Response to Reviewers:

Thank you for your valuable feedback which has helped enhancing the clarity and precision of our study and for acknowledging that the previous comments have been addressed sufficiently.

Review comment to the Author:

Reviewer #3: My previous comments have been sufficiently addressed. Some things to add would be to state which version of Pfizer Biontech vaccine was implemented and considered during study time as many different updated vaccines emerged. Some of those were more effective against omicron than others. Similarly, please clarify what is meant by “fully vaccinated” in the “Definition of vaccination status:” section highlighting which vaccine version was considered. The originally implemented ones? If so please clarify accordingly.

I agree with the statement in the discussion which states: Key findings of the study indicate that the Pfizer-BioNTech vaccine is effective in reducing the odds of developing symptomatic COVID-19. This should have been highlighted more and emphasized in the abstract as it is an important finding. The abstract reads almost as vaccination not having an enough positive impact where the authors' key findings clearly indicates otherwise. Please update accordingly.

Response to reviewer #3:

Response to Reviewer #3

Regarding your additional suggestions:

1. Pfizer-BioNTech Vaccine Version:

We appreciate your suggestion to specify the version of the Pfizer-BioNTech vaccine implemented during the study. During the study period, the original Pfizer-BioNTech COVID-19 vaccine (BNT162b2) was the version available and administered in Lebanon. We have updated the manuscript to clarify this point.

2. Definition of “Fully Vaccinated”:

We have clarified the definition of “fully vaccinated” in the “Definition of vaccination status” section. In the context of this study, “fully vaccinated” refers to individuals who received the complete initial series of the original Pfizer-BioNTech COVID-19 vaccine (two doses of BNT162b2). This definition has been adjusted in the manuscript to avoid any ambiguity.

3. Abstract adjustments:

We have revised the abstract to state the protective effects of vaccination, including its role in mitigating symptom severity.

Thank you again for your thoughtful remarks.

Dear editors,

Thank you for your feedback. We have reviewed the reference list and made the necessary updates and adjustments as per your recommendations. The following changes have been made:

1. Reference 2: The link to the Centers for Disease Control and Prevention (CDC) website has been updated as follows:

o Centers for Disease Control and Prevention. (n.d.). Ncov 2019 – Variants classifications. Available from: https://www.cdc.gov/covid/?CDC_AAref_Val=https://www.cdc.gov/coronavirus/2019-ncov/variants/variant-classifications.html (visited 20-9-2022).

2. References 5 and 13: Typographical errors were corrected as follows:

o Tan, S. H., Cook, A. R., Heng, D., Ong, B., Lye, D. C., Tan, K. B., et al. (2022). Effectiveness of BNT162b2 vaccine against Omicron in children 5 to 11 years of age. New England Journal of Medicine, 387(6), 525–532. https://doi.org/10.1056/NEJMoa2203209.

o Li, Q. H., Ma, Y. H., Wang, N., Hu, Y., & Liu, Z. Z. (2020). New coronavirus-infected pneumonia Engulfs Wuhan. Asian Toxicology Research, 1, 1–7.

Best Regards,

---

## [Editor Report · Decision Letter 2]

15 Jan 2025

Effectiveness of the Pfizer-BioNTech Vaccine against the Omicron Variant of SARS-CoV-2 among Adults Aged 50 and Above: A Case-Control Study in Lebanon, June 2022

PONE-D-24-09717R2

Dear Dr. Baakliny,

We’re pleased to inform you that your manuscript has been judged scientifically suitable for publication and will be formally accepted for publication once it meets all outstanding technical requirements.

Kind regards,

Marwan Osman, PhD, MPH

Academic Editor

PLOS ONE

---

## [Editor Report · Acceptance letter]

PONE-D-24-09717R2

PLOS ONE

Dear Dr. Baakliny,

I'm pleased to inform you that your manuscript has been deemed suitable for publication in PLOS ONE. Congratulations! Your manuscript is now being handed over to our production team.

Kind regards,

on behalf of

Dr. Marwan Osman

Academic Editor

PLOS ONE